

# Carbon monoxide air-pollution on sub-city scales and along arterial roads detected by the Tropospheric Monitoring Instrument

Tobias Borsdorff[1], Joost aan de Brugh[1], Sudhanshu Pandey[1], Otto Hasekamp[1], Ilse Aben[1], Sander Houweling[1,2], and Jochen Landgraf[1]

[1]SRON Netherlands Institute for Space Research, Utrecht, the Netherlands
[2]Department of Earth Sciences, Vrije Universiteit, Amsterdam, the Netherlands

**Correspondence:** Tobias Borsdorff (t.borsdorff@sron.nl)

**Abstract.** The Tropospheric Monitoring Instrument (TROPOMI) on the Sentinel-5 Precursor satellite provides measurements of carbon monoxide (CO) total column concentrations based on Earthshine radiance measurements in the 2.3 $\mu$m spectral range with a spatial resolution of $7\times7$ km$^2$ and daily global coverage. Due to the high accuracy of the observations, CO pollution can be detected over cities and industrial areas using single orbit overpasses. In this study, we analysed local CO enhancements

in an area around Iran from 1 November to 20 December 2017. We employed the Weather Research and Forecasting (WRF) model v3.8.1 using the EDGAR v4.2 emission inventory and evaluated CO emissions from the cities of Tehran, Yerevan, Urmia, and Tabriz on a spatial resolution comparable to that of TROPOMI. At large scales, TROPOMI CO agrees well with the WRF simulation with a mean difference of 5.7 %. However, the emissions for the city area had to be significantly increased in order to match the observations. Moreover, significant differences at sub-city scale remain. To match the TROPOMI CO

observations around the Armenian city of Yerevan, it is necessary to introduce CO emissions along a southeast arterial road of Yerevan. Overall, this hints at deficits in the EDGAR inventory in the region around Iran and indicates TROPOMI's capability to identify localised CO pollution on sub-city scales, which at the same time challenges current atmospheric modelling at high spatial and temporal resolution.

## 1   Introduction

The Tropospheric Monitoring Instrument (TROPOMI) was launched on the Copernicus Sentinel-5 Precursor satellite since 13 October 2017. The instrument is a nadir looking pushbroom grating spectrometer, that performs measurements of the solar light reflected by the Earth's atmosphere in the UV-VIS (270-495 nm), NIR (710-775 nm), and SWIR (2305-2385 nm) spectral domain. The novelty of the mission is the combination of high spatial resolution of the measurements ($7 \times 7$ km$^2$, SWIR) in nadir observation geometry, the daily global coverage and a high signal-to-noise ratio (Veefkind et al., 2012).

One of the primary objectives of the mission is to measure the total column concentration of carbon monoxide (CO), which is an atmospheric trace gas emitted to the atmosphere mainly by incomplete combustion. The major sink of CO is its reaction with the hydroxyl radical OH (Spivakovsky et al., 2000). With its low background concentration (ca. 80 parts per billion (ppb) at the Northern Hemisphere) and the moderately long atmospheric residence time of weeks to months (Holloway et al., 2000),



CO is a good tracer to monitor atmospheric transport processes (Gloudemans et al., 2009) as well as to detect pollution sources of natural (e.g. biomass burning, wild fires; Yurganov et al. (2004, 2005)) and anthropogenic origin (e.g. mega cities; Pommier et al. (2013); Stremme et al. (2013)).

Landgraf et al. (2016a, b) developed the operational code for the TROPOMI mission to retrieve the total column con-
centration of CO from the SWIR measurements of the instrument. The high spatial resolution in combination with a high signal-to-noise ratio of the measurements allows the detection of CO pollution from large cities (e.g. Mexico, Tehran, Isfahan) and industrial areas (e.g. Po valley in Italy) from single orbit overpasses and to track the transport of pollution on regional (e.g. India) to global scales (e.g. biomass burning in Africa) with daily global coverage (Borsdorff et al., 2018b). Borsdorff et al. (2018a) reported a good agreement between the TROPOMI CO dataset and the simulations of the ECMWF-IFS model.
Already in the early phase of the mission a validation with TCCON and NDACC ground-based measurements showed compliance with the mission objectives on precision and accuracy (Borsdorff et al., 2018b). In July 2018, the dataset was released to the public (ESA, 2018).

For this study, we analyse TROPOMI observations using CO tracer simulations for the period 1 November - 20 December 2017 using the Weather Research and Forecasting model (WRF) (Skamarock et al., 2008). The model domain is centred over
Iran with a spatial resolution comparable to that of TROPOMI. During this period frequent clear sky observations are possible over this region, which makes it particularly suited to study localised hot spots of CO emissions from urban areas. Considering the CO emissions from the cities of Yerevan, Urmia, Tabriz, and Tehran as independent atmospheric tracers, we compared the modelled CO columns with spatio-temporally coincident TROPOMI measurements to evaluate TROPOMI's monitoring capability of CO emissions on city and sub-city scales.
The paper is structured as follows: Section 2 introduces the TROPOMI CO dataset and the WRF model simulations. Section 3 presents our approach to estimate emissions from the TROPOMI data for the example of the city of Tehran and a selected domain over Armenia. Finally, Sec. 4 summarises and concludes our study.

## 2  Data sets

### 2.1  TROPOMI CO total column densities

The TROPOMI CO dataset is inferred from the 2.3 $\mu$m measurements of the instrument by deploying the shortwave infrared retrieval algorithm SICOR, which was developed for the operational processing of TROPOMI data. The algorithm is based on the profile scaling approach, whereby a prior vertical distribution of CO is scaled to fit the observation (Borsdorff et al., 2014). The implementation and retrieval settings are presented in detail by Landgraf et al. (2016a), where the CO profile, to be scaled by the retrieval, is taken from monthly averaged simulations of the global chemical transport model TM5 (Krol et al.,
2005) with a latitude/longitude resolution of $3^o \times 2^o$. The retrieval accounts for atmospheric light scattering by clouds and aerosols and estimates the trace gas columns together with surface albedo and effective cloud parameters (cloud height ($z$) and cloud optical thickness ($\tau$)) to account for the cloud contamination of the measurements (Landgraf et al., 2016b). An essential element of the TROPOMI CO data product is the column averaging kernel $\mathbf{A}_{\mathrm{col}}$, which describes the sensitivity of the retrieved



CO column $c_{\mathrm{ret}}$ to changes in the true vertical profile $\boldsymbol{\rho}_{\mathrm{true}}$ of CO (Rodgers, 2000), namely

$$c_{\mathrm{ret}} = \mathbf{A}_{\mathrm{col}}\boldsymbol{\rho}_{\mathrm{true}} + \epsilon_{\mathrm{CO}} \,, \tag{1}$$

where $\epsilon_{\mathrm{CO}}$ represents the error of the retrieved CO column.

Borsdorff et al. (2018b, a) showed the validity of the CO data product for both clear sky and cloudy measurement conditions.
This study considers only TROPOMI clear sky observations ($\tau < 0.5$ and $z < 5$ km, over land) with good sensitivity to CO in the tropospheric boundary layer close to the emission sources. This data filtering is described in more detail by Borsdorff et al. (2018a).

## 2.2   WRF model simulations

For a domain of 2408 x 2674 km$^2$ around Iran centred at 50.2$^o$E and 33.5$^o$N, we simulated the atmospheric CO field using
WRF v3.8.1 (Skamarock et al., 2008) and its CHEM module by Grell et al. (2005) for tracer transport. Photochemical oxidation and secondary production of CO in the atmosphere have been ignored, justified by the long lifetime of CO compared with the size of the model domain (Dekker et al., 2017, 2018). Figure 1 shows the topography of the area based on the 2000 Shuttle Radar Topography Mission (SRTM) data with an resolution of 15 arcsec (Farr et al., 2007). For the cities of Yerevan, Urmia, Tabriz, and Tehran, we have identified pollution hot spots in the TROPOMI data, as will be discussed in the next section. The
CO simulations were performed for the period 1 November to 20 December 2017 at a resolution of $7 \times 7$ km$^2$ and at 29 pressure levels from the Earth's surface up to 50 hPa. The WRF settings were the same as used in Dekker et al. (2017), including the Yonsei University (YSU) boundary layer scheme (Hu et al., 2013), and the convection parameterisation by Grell and Freitas (2014). The initial and boundary conditions of CO are adapted from ECMWF-CAMS near real-time analysis data (George et al., 2015). The WRF simulation has been nudged to NCEP final analysis meteorological fields (ds083.2, NCEP (2000)) at
$1^o$x$1^o$ and 6 hourly resolution at the model's initial and domain boundaries. Anthropogenic surface emissions of CO for 2010 are from the EDGARv4.2 emission inventory (Crippa et al., 2016).

To disentangle the CO emissions from different urban areas, we isolated the EDGAR CO emissions of several cities from the remaining emissions and treated them as individual tracers. Figure 2 shows the selected city emissions from the EDGAR inventory in a 70 km radius around Tehran and a 30 km radius around the cities of Yerevan, Tabriz and Urmia. Alternatively,
the Yerevan emissions can be replaced by a spatially extended emission source covering the city including an arterial road. The strengths of all these city emission sources in our spatial domain are summarised in Table 1.

Assuming the linearity of the simulated CO concentration with respect to the source strength of a tracer, we can express the total CO columns of the WRF simulation by a superposition of the individual tracers, namely

$$\mathrm{CO}_{\mathrm{total}} = \boldsymbol{\alpha}^T \boldsymbol{C} \tag{2}$$

with

$$
\begin{aligned}
\boldsymbol{\alpha} &= \ (\alpha_{\mathrm{bkg}},\ \alpha_{\mathrm{Tehran}},\ \alpha_{\mathrm{Urmia}},\ \alpha_{\mathrm{Tabriz}},\ \alpha_{\mathrm{Yerevan/road}},\ \alpha_{\mathrm{rest}}) \\
\boldsymbol{C} &= \ (\mathrm{CO}_{\mathrm{bkg}},\ \mathrm{CO}_{\mathrm{Tehran}},\ \mathrm{CO}_{\mathrm{Urmia}},\ \mathrm{CO}_{\mathrm{Tabriz}},\ \mathrm{CO}_{\mathrm{Yerevan/road}},\mathrm{CO}_{\mathrm{rest}})
\end{aligned}
\tag{3}
$$





where the different coefficients of vector $\boldsymbol{\alpha}$ correspond to a scaling of the prior CO sources. The CO fields from the individual hot spots, $CO_{bkg}$ denotes the background field originating from the ECMWF-CAMS boundary conditions and $CO_{rest}$ summarises the contribution of the remaining EDGAR sources. Thus, the coefficients $\boldsymbol{\alpha}$ in Eq. (2) can be adjusted to fit the TROPOMI CO observations in the spatial domain of interest.

## 3 Data analysis

### 3.1 Scaling of the EDGAR emissions

To compare the simulated CO fields with the TROPOMI CO column, we first interpolated the WRF data to the TROPOMI observation in time and space. Subsequently, by analogy with Eq. (1), we applied the total column averaging kernel $\mathbf{A}_{col}$ of the TROPOMI CO product to the corresponding model profile. In this way, we account for the CO column sensitivity of the retrieval in our comparison (Borsdorff et al., 2014).

We start with a WRF model run using the EDGAR emission without any adjustments. Figure 3 shows the comparison of the TROPOMI CO data with the collocated CO field of the WRF simulation for an overpass on 18 December 2017. The overall agreement is good with a mean difference of 5.7 %, which agrees well with the finding of Borsdorff et al. (2018b); who compared the TROPOMI CO data with ECMWF-CAMS near-real-time analysis data. This is expected because the WRF simulation is constrained by the same data at the domain boundaries. However, a closer look reveals that the model underestimates the CO enhancements sensed by TROPOMI above pollution hot spots, hinting at an inconsistency between the observations and the EDGAR emissions over urban areas. Furthermore, large scale deviations occur in the North-East of the model domain that are most probably caused by the ECMWF-CAMS side constraints.

Having the different tracer fields available, we fitted the coefficients $\boldsymbol{\alpha}$ shown in Eq. 2 with a standard least squares method to improve the match between the simulated CO fields and the TROPOMI observations. Thus, this optimisation yields the relative change of the emission for the different tracer sources, where we estimate the corresponding uncertainties by bootstrapping. In this, we created 1000 data samples by repeatedly reducing the TROPOMI data to 50% of the original data volume. Subsequently for each of those samples we estimated the emission sources with the approach described above. The statistics of the results are shown in Table 1 were we report the mean and the standard deviation of the ensemble as a robust error estimate. The fitting of the tracer fields improves the agreement between TROPOMI and WRF; however we still see significant differences on sub-city scales, and missing emissions in the EDGAR inventory hamper the interpretation of the TROPOMI data. These points will be discussed in the following sections.

### 3.2 Sensing pollution on sub-city scales in Tehran

The city of Tehran shows clearly enhanced CO concentrations well isolated from the low surrounding background concentration of about 80 ppb that is pronounced in many TROPOMI overpasses over Tehran. Figure 4 compares the TROPOMI CO measurements with the WRF simulation over Tehran for 18 December 2017. The WRF data (top right panel) were fitted to the



TROPOMI measurements (top left panel) by scaling WRF's background CO field with $\alpha_{\text{bkg}} = 0.91 \pm 0.016$ and the Tehran emissions with $\alpha_{\text{Tehran}} = 2.24 \pm 0.24$. The other CO tracer fields of WRF were kept unchanged. The remaining differences show that overall the simulation and the TROPOMI data agree well but with significant differences across the city. Although the model has a high CO sensitivity at the centre of Tehran (lower right panel) a trustworthy emission estimate would only be

possible if the emission inventory gave a more realistic spatial distribution of pollution sources within the hot spot. Modelling CO on sub-city scales is challenging for the WRF model and the doubling of the EDGAR emissions for the city of Tehran that is inferred from the TROPOMI measurements must be considered with cautiousness.

On 18 December 2017 the pollution of Tehran was transported up to 600 km eastwards and the corresponding CO plume is nicely reflected in both the TROPOMI data and the WRF simulation; as shown in Figure 5. Fitting the model to the

TROPOMI data over the entire plume domain results in a scaling of the EDGAR Tehran emissions by $\alpha_{\text{bkg}} = 0.97 \pm 0.006$ and $\alpha_{\text{Tehran}} = 1.48 \pm 0.14$ but when leaving out the urban area of Tehran, the source inversion results in a scaling of $\alpha_{\text{bkg}} = 0.99 \pm 0.001, \alpha_{\text{Tehran}} = 0.86 \pm 0.03$ of the EDGAR emissions. The inconsistency of the emission estimates may come from an inappropriate simulation of the hot spot as mentioned above but may also be caused by temporal variability of the emission at Tehran or small errors in the wind fields thereby causing shifts in the downwind area. Based on the EDGAR inventory, the

WRF simulations assume time invariant emissions. However, the transport of 600 km needs about 33 hours assuming a wind speed of about 5 ms$^{-1}$. Hence, the plume could reflect a different source strength that changed through time.

### 3.3   Sensing pollution along main traffic roads

On 27 November and 12 and 17 December 2017, TROPOMI detected a sequence of strong CO pollution events near to the city of Yerevan in Armenia, which are depicted in Fig. 6. The enhancements in CO show a clear time dependency, probably

due to varying local meteorology. The high CO concentration follows the orographic pattern of the region in Fig. 3 and indicate an accumulation of pollution in the mountainous region. Furthermore, CO pollution hot spots at the cities of Urmia and Tabriz are clearly visible. Here, the WRF tracer transport simulations of CO are used to conclude on the strength of the emission sources and how local winds impact the strong CO enhancement sensed by TROPOMI. Figure 7 compares the 17 December pollution event measured by TROPOMI with corresponding WRF simulation. Using the EDGAR emissions,

the large differences indicate that the enhancement cannot be explained by atmospheric transport as simulated by WRF and indicates an underestimation of the CO emissions in the EDGAR data.

The two middle panels of Fig. 7 show the situation after adjusting the urban emission strengths by the optimized scaling factors $\alpha_{\text{Yerevan}}$, $\alpha_{\text{Tabriz}}$, and $\alpha_{\text{Urmia}}$, and these emissions are summarised in the first three rows of Table 2. Here, the emissions of the remaining tracers $CO_{\text{Tehran}}$ and $CO_{\text{rest}}$ were not adjusted. The results indicate that the prior assumed emissions are too

low for Tabriz and far too low for Yerevan and Urmia. After adjusting the emission sources, the WRF simulation can explain to a major extent the observed regional CO enhancement. It is remarkable how well the WRF model describes the pollution at Tabriz and Urmia and the nearby valley. Also, the accumulation of CO pollution from Yerevan westwards of the city is well reproduced by the model, but still a large differences remains south-east of the city.





To account for the residuals, we extended the emission pattern of Yerevan along a main traffic roadway of the city, as indicated in Fig. 2. After fitting the emissions, the agreement between the TROPOMI observations and WRF simulations is clearly improved (see right panels of Fig. 7). The estimated emissions are shown in the last three rows of Table 2. The improved fit convincingly shows the need for a more extended emission source along the arterial road of the city, which is not represented in the current EDGAR inventory but could be identified with TROPOMI CO measurements. Obviously, the CO measurements cannot attribute missing emissions to specific processes. So both high traffic load and/or other activities e.g. industrial, along the road may explain our findings. Table 2 also summarises the emission estimates for the other days, 27 November and 12 December, which indicates emission of similar magnitude but with some temporal variability. Follow-up studies must show if this variability can be attributed to a variation of the emission sources or to biases of the inversion approach used. Furthermore, looking at the result for individual cities, in this area, we recognize remaining differences after optimisation of the emissions. As discussed for Tehran in the previous subsection, we assume that TROPOMI is sensitive to pollution patterns at sub-city scale which are not well reflected by the EDGAR data set.

## 4   Conclusions

In this study, we compared TROPOMI CO total column densities with CO tracer simulations of the WRF model v3.8.1 for a regional domain centred over Iran from 1 November to 20 December 2017 with a spatial resolution comparable to that of TROPOMI ($7 \times 7$ km$^2$). Anthropogenic surface emissions of the simulations are based on the EDGARv4.2 emission inventory. Considering the CO emissions from the cities of Yerevan, Tehran, Urmia, and Tabriz as independent atmospheric tracers allowed us to infer the strength of CO surface emissions from the observed CO enhancements in the TROPOMI data. For comparing TROPOMI CO data with model simulations, it was necessary to bring both data sets to the same spatio-temporal sampling and to apply column averaging column of the TROPOMI product to the model data even though this introduces computationally expensive operations.

When looking over the full domain, we found the WRF simulations using EDGAR emissions are in good agreement with the TROPOMI measurements with a mean difference of 5.7 percent. Both TROPOMI and WRF show the same large scale variation of CO over the region considered. A CO pollution plume seen by TROPOMI with an extension of 600 km and its origin at Tehran could be simulated well by the WRF model considering synoptic transport of CO emissions from the city. On city scales we identified significant differences between model simulation and TROPOMI observations, which we attribute to shortcomings in the EDGAR inventory due to time invariant emissions, underestimation of existing emission sources and lack of sufficient emission sources in Armenia. Also, the capability of the WRF model to simulate CO on sub-city scales might be an additional source of uncertainty.

For Tehran, we show that the spatial emission patterns of the data set are not consistent with the TROPOMI observations, hinting at problems of the emission inventory at sub-city scales. The strong CO enhancements over Yerevan, Urmia and Tabriz seen by TROPOMI are only reflected by the model after substantially adjusting the emission strength of the city emissions.





Additionally, a series of pollution events near Yerevan can only be reproduced by WRF simulations when CO emissions along an arterial road are introduced in addition to the EDGARv4.2 inventory.

In this study, we showed that the TROPOMI CO dataset is capable of distinguishing CO pollution on city and sub-city scales and even can detect CO pollution along an arterial road. We demonstrated that TROPOMI can identify new emission sources and can thereby help to fill gaps in emission inventories, which at the same time challenges atmospheric modelling on the spatial and temporal scales observed by TROPOMI.

*Data availability.* The TROPOMI CO data set and the WRF simulations of this study are available for download at ftp://ftp.sron.nl/ pub/pub/DataProducts/TROPOMI_CO/. The underlying data of the figures presented in this publication can be found at ftp://ftp.sron.nl/ open-access-data/.

*Author contributions.* Tobias Borsdorff, Joost aan de Brugh, Otto Hasekamp and Jochen Landgraf provided the TROPOMI CO retrieval and data analysis. Sudhanshu Pandey, Ilse Aben and Sander Houweling are responsible for the WRF simulation. Jochen Landgraf supervised the study. All authors discussed the results and commented on the manuscript.

*Competing interests.* The authors declare no competing interests.

*Disclaimer.* The presented work has been performed in the frame of the Sentinel-5 Precursor Validation Team (S5PVT) or Level 1/Level 2 Product Working Group activities. Results are based on preliminary (not fully calibrated/validated) Sentinel-5 Precursor data that might change in the future.

*Acknowledgements.* We would like to thank the team that has realised the TROPOMI instrument, consisting of the partnership between Airbus Defense and Space, KNMI, SRON and TNO, and commissioned by the Netherlands Space Office (NSO) and the European Space Agency (ESA). Sentinel-5 Precursor is a ESA mission on behalf of the European Commission (EC). The TROPOMI payload is a joint development by ESA and the NSO. The Sentinel-5 Precursor ground-segment development has been funded by ESA and with national contributions from The Netherlands, Germany, and Belgium. This research has been funded in part by the TROPOMI national program from the NSO. The TROPOMI data processing was carried out on the Dutch national e-infrastructure with the support of the SURF Cooperative. The work contains modified Copernicus Atmosphere Monitoring Service Information [2017]. Neither the European Commission nor ECMWF is responsible for any use that may be made of the information it contains.



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





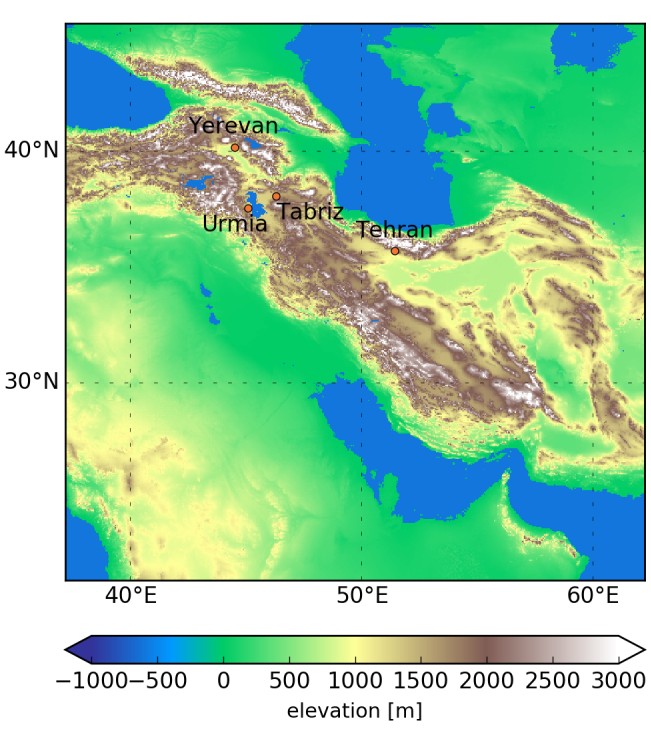

**Figure 1.** Topological map of the domain used for the WRF simulation with the cities of Yerevan, Urmia, Tabriz, and Tehran marked.



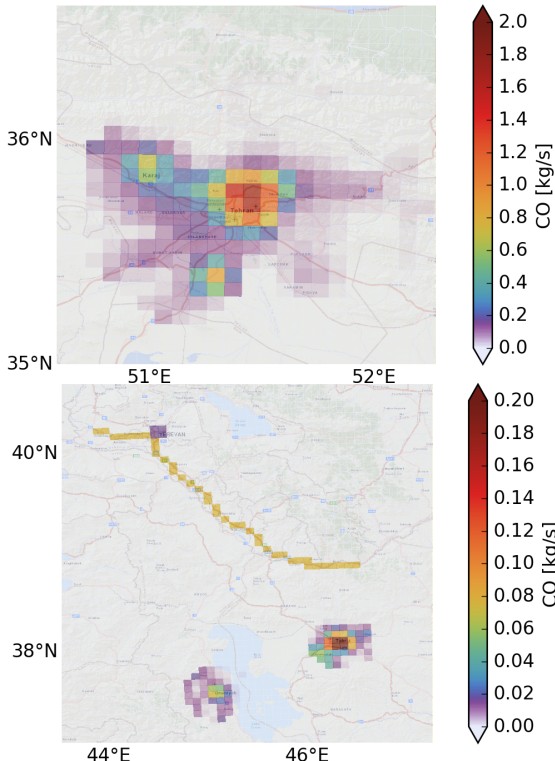

**Figure 2.** Emission used for the WRF tracer runs. Emission 70 km around Tehran (upper panel) extracted from EDGAR. (lower panel) The emissions of Yerevan, Tabriz, and Urmia extracted from EDGAR 30 km around the cities, and the additionally introduced emission along a main traffic road in Armenia not present in EDGAR.

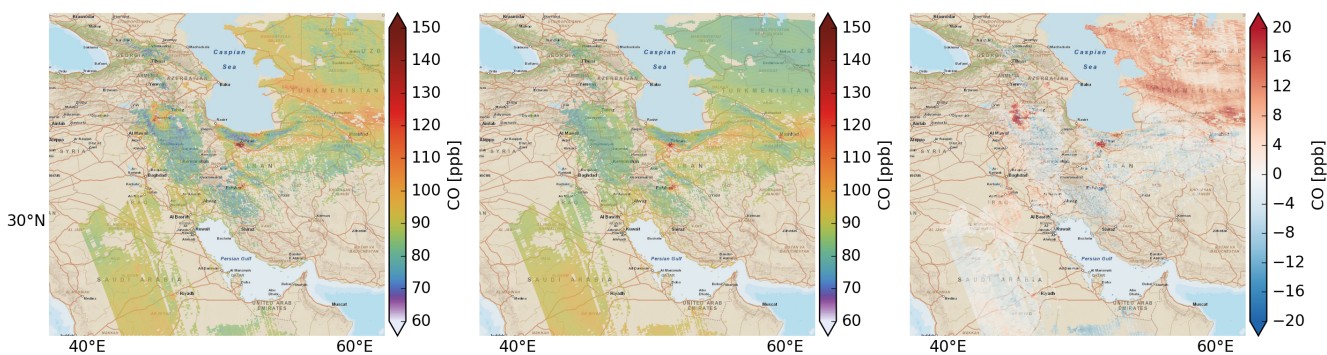

**Figure 3.** TROPOMI CO retrieval of one orbit on 18th December 2017 (left panel), the collocated WRF simulation scaled to the TROPOMI data (middle panel), and the difference (TROPOMI - WRF, right panel).



**Figure 4.** TROPOMI CO retrieval above Tehran on the 18th December 2017 (top left panel), The collocated WRF simulation scaled to the TROPOMI data (top right panel), the difference between the TROPOMI and scaled WRF data (TROPOMI - WRF, lower left panel), and the sensitivity of fitting the WRF data to the TROPOMI retrieval (lower right panel).



**Figure 5.** Same as Fig. 4 but a lager domain was used to include the pollution plume of Tehran in eastward direction. From top to bottom: TROPOMI, WRF, (TROPOMI-WRF), and sensitivity of fitting the WRF data to the TROPOMI retrieval.



**Figure 6.** Sequence of high CO pollution events measured by TROPOMI on the 27th (top panel) November, 12th (middle panel), and 17th December 2017 (lower panel).





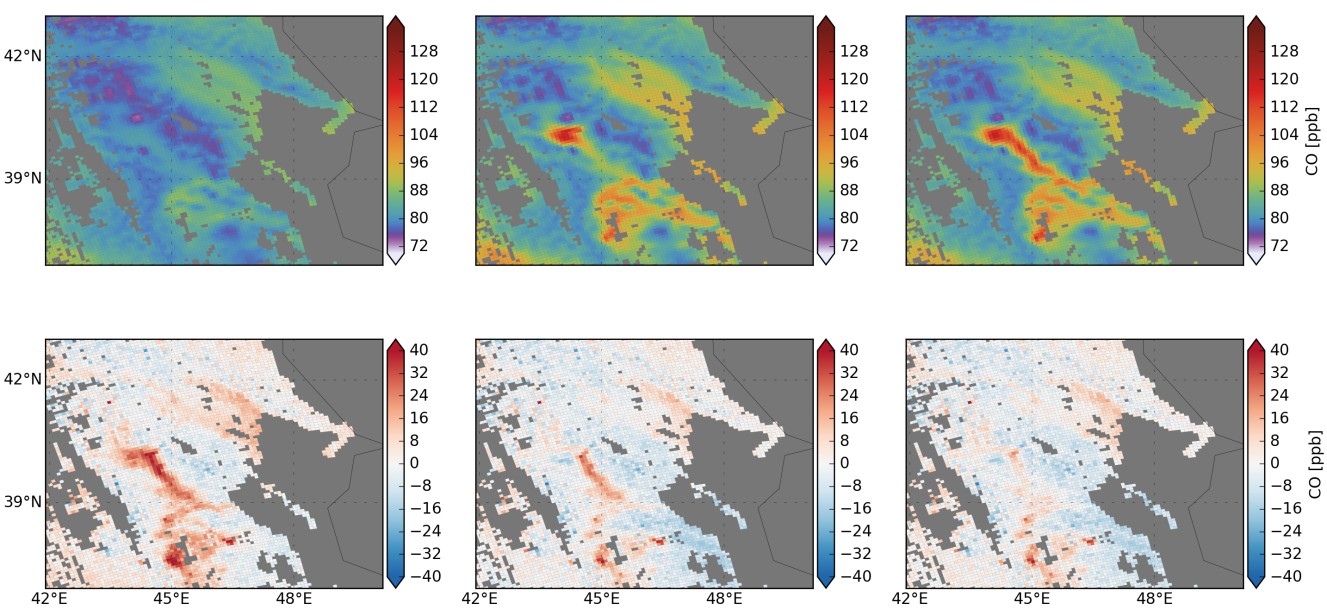

**Figure 7.** Comparison of TROPOMI CO and WRF (top row: WRF simulation, lower row difference TROPOMI - WRF). Left column: WRF run with EDGAR emissions. Middle column: WRF run with adjusted emissions for Yerevan, Tabriz and Urmia. Right column: WRF run with adjusted emission of Yerevan, Tabriz and Urmia and the traffic route near Yerevan.





**Table 1.** CO emissions extracted from the EDGAR inventory in a radius around the indicated cities and inverted from the TROPOMI CO retrievals shown in Fig. 3. on 18th December 2017

| name | radius [km] | latitude [degree] | longitude [degree] | EDGAR CO emission [kg/s] | inverted CO emission [kg/s] |
|---|---|---|---|---|---|
| Yerevan | 30 | 40.18 | 44.51 | 0.07 | $31.24 \pm 2.04$ |
| Urmia | 30 | 37.55 | 45.08 | 0.64 | $11.73 \pm 2.10$ |
| Tabriz | 30 | 38.06 | 46.31 | 2.29 | $10.73 \pm 1.49$ |
| Tehran | 70 | 35.70 | 51.42 | 33.72 | $50.58 \pm 3.14$ |

**Table 2.** Emissions [kg/s] estimated from the TROPOMI measurements shown in Fig. 6. First three rows when fitting the emission of Yerevan, Tabriz and Urmia, and last three rows when fitting the emission of Tabriz, Urmia, and an assumed pollution along a traffic route near to Yerevan.

| name | 27 Nov. 2017 | 12 Dec. 2017 | 17 Dec. 2017 |
|---|---|---|---|
| Yerevan | $10.42 \pm 1.34$ | $13.56 \pm 0.93$ | $7.75 \pm 0.50$ |
| Urmia | $7.90 \pm 2.60$ | $4.93 \pm 1.65$ | $11.23 \pm 2.58$ |
| Tabriz | $0.05 \pm 2.06$ | $2.64 \pm 1.40$ | $-4.17 \pm 2.31$ |
| Yerevan/road | $16.57 \pm 0.81$ | $25.46 \pm 1.03$ | $22.50 \pm 0.74$ |
| Urmia | $9.30 \pm 2.60$ | $6.77 \pm 1.61$ | $14.31 \pm 2.25$ |
| Tabriz | $0.36 \pm 2.09$ | $3.31 \pm 1.38$ | $0.62 \pm 1.95$ |