# Peer review of "Carbon monoxide air-pollution on sub-city scales and along arterial roads detected by the Tropospheric Monitoring Instrument"

_Atmospheric Chemistry and Physics, 2018_

## Referee Comment (RC1) · Anonymous Referee #1 · 28 Dec 2018

General comments

The manuscript presents an excellent example of an application of the use of TROPOMI data for air pollution in cities as well as in roads. A comparison between TROPOMI CO observations and the WRF model using the EDGAR v4.2 emission inventory is explored and differences at large scales and "city area" scales are evaluated, finding large differences at smaller (city area) scales.

Specific comments

On Page 5, line 5, the authors address the importance of a "more realistic spatial distribution of pollution sources within the hot spot". Could the authors elaborate some

more about this issue?

As this study is a highly valuable example for air pollution studies, the authors could provide further considerations while conducting a similar experiment in other cities, areas or countries and at different scales (city and sub-city).

Technical corrections

Page 5, line 33: difference Page 6, line 20: remove column

———————————————————

---

## Referee Comment (RC2) · Anonymous Referee #2 · 11 Jan 2019

**General comments**

The paper compares carbon monoxide (CO) total column concentrations from TROPOMI with CO column concentrations derived from a WRF simulation including CO as a passive tracer for Northern Iran and Armenia. The analysis indicates that CO emissions from the EDGAR inventory, which were used as input for the WRF simulations, are be too low for urban areas and that emission from major roads are missing. The paper presents a nice example of an application of the use of TROPOMI data for model evaluation. However, it suffers from a somewhat too brief description of the research and the lack of details. Adding a more in-depth description of the applied

methods and some more discussion would make the paper much more useful and also better readable.

**Detailed comments (including also minor points)**

P 1, l 8: 'At larger scales' = 'For background conditions'?

P 1, l 15: 'since 13 October' = 'on 13 October'?

P 2, l 10: Please explain TCCON and NDACC.

P 3, l 1: What is the 'true vertical profile' in the context of this paper?

P3, l 11-20: Please give some more details of the model setup. For example, which chemistry and/or tracer option is applied? How many tracers are considered? Which model resolution is applied? Why does the model domain extend that far towards the South?

P 3, l 13: This particular topography is certainly not applied for the model simulations. So, for which purpose is it shown here?

P 3, l 21: Please add some more information about the EDGAR emissions and how they are applied here.

P 3, l22-32: The first line of the paragraph seems to be misplaced (or the second and third sentence should be part of the part of the description of the EDGAR emissions).

P 3, l 27-32: At this point it is not clear why this is part of the description of WRF. Please explain this in the context with the WRF tracers.

P 4, l 1-2: Something seems to be missing in this sentence, please reword.

P4, l 1: Please explain the meaning of 'prior CO sources'.

P 4, l 7-8: Does the interpolation of the WRF output in time and space mean that e.g. the top right of Figure 4 is a combination of outputs at different output times. If so, please mention this and/or give a description how this interpolation was made.

[Figure]

P 4, l 11 and l 31: at which time(s)?

P 4, l 17-18: This could be determined easily by inspection of the EMWF-CAMS fields.

P 4, l 24: Which ensemble?

P 4, l 13: Since emissions are usually higher during the daytime than during the night, the error due to temporally invariant emissions may depend on the time of the day. Therefore, information on the considered time of the day may be interesting.

P 5, l 15: This should also be mentioned already in section 2.2.

P 5, l 18-19: To what extent did these pollution events show up in the WRF simulations when the original emissions were applied?

P 5, l 31: Does the WRF simulation really 'explain' the observed CO enhancement?

P 6, l 29: The sentence 'The poor capability . . .' is incomprehensible. Why is WRF not able to simulate at city scale. Is it due to the resolution, due to the emissions, or due to something else?

P 7, l 2: What happens if the adapted emissions are applied for the entire episode? Is CO overestimated during observed episodes with moderate CO, or are the results still ok?

P 7, l 5: Please add some sentences on the potential of the method for other regions of the world.

Data availability: The first link does not work and the second one is not really helpful in its current form. Please correct the links and add some explanation (if necessary, add supplementary material).

Figure 3 includes numerous details, which are not necessary or enlarge the figure. Please mention the hour or time interval of the shown orbit.

Figure 4 and Figure 5: Please mention the hour, time interval or orbit(s) in the figure

caption.

Caption of Fig. 6: Please mention the region shown in this figure also in the caption.

Figures 6: Does this figure display the same region as the lower part of Fig. 2? If not: why? Please mention the time interval in the figure caption.

Caption of Fig. 7: Please mention the date and the time (interval).

---

## Referee Comment (RC3) · Anonymous Referee #3 · 16 Jan 2019

The authors have used TROPOMI CO column data over Iran and Armenia to estimate CO emissions from Tehran, Yerevan, Urmia, and Tabriz between 1 November – 20 December 2017. As a result of its high accuracy and observational coverage, TROPOMI is able to capture the influence of urban emissions at unprecedented temporal and spatial resolution. Using WRF simulations, the authors found that the EDGAR v4.2 inventory significantly underestimates CO emissions from Yerevan, Urmia, Tabriz, and Tehran. Furthermore, they found that the TROPOMI data suggested transportation emissions in the vicinity of Yerevan that are absent in the EDGAR v4.2 inventory. The manuscript nicely shows the potential of the TROPOMI data for improving our knowledge of urban scale CO emissions. I agree with the authors that the data will challenge

our current modeling approaches at high spatiotemporal resolution. I therefore recommend the manuscript for publication in ACP after the authors have addressed my comments below.

General Comments

1) It would be useful to see what the background CO fields look like across the region. How much do they contribute to the total columns shown in Figure 3, for example? How much does the background contribute to the variability in the model simulations of the observations on 27 November, 12 December, and 17 December? Knowing how the background is changing on these days will help interpret the variations in the estimated emissions (shown in Table 2) on these days for Yerevan, Urmia, and Tabriz.

2) The analysis produced negative emissions of -4.17 kg/s, with an uncertainty of 2.31 kg/s, for Tabriz on December 17th. How do the emissions go from 2.64 kg/s on Dec 12 to -4.17 kg/s on Dec 17th? I realize that the authors stated, concerning future work, that "follow-up studies must show if this variability can be attributed to a variation of the emission sources or to biases of the inversion approach used," but how does one explain this sink of CO on Dec 17th in the context of urban sources of CO? It would be helpful to learn more about the fitting process for the emissions. Also, could discrepancies in accounting for the background contribute to this negative estimate for the emissions?

3) How is the sensitivity that is shown in Figures 4 and 5 calculated? Also, how does the sensitivity for Yerevan, Urmia, and Tabriz vary between Nov 27th, Dec 12th, and Dec 17th?

4) Is the sensitivity to emissions from Urmia and Tabriz as localized as that for Tehran? I ask because Table 2 shows that including the road emissions produces large changes in the emissions for Urmia and Tabriz, even though these two cities are to the south of the region of the road emissions (as shown in Figure 2). If the fitting sensitivity for these cities is also localized, why are the emissions changing so much when the road

emissions are included?

Technical Comments

1) Page 1, lines 7-8: Since the WRF simulation is not being used to evaluate the TROPOMI data, I would suggest changing the order of this sentence to: "The WRF simulation agree well with TROPOMI CO, with a mean difference of 5.7%."

2) Page 1, lines 15-16: Please change "since 13 October 2017" to "on 13 October 2017."

3) Page 1, line 23: Please change "at the Northern Hemisphere" to "in the Northern Hemisphere."

4) Page 2, line 10: Please define TCCON and NDACC.

5) Page 2, line 11: Can you please state what are the precision and accuracy requirements? It would help the reader with interpreting the results of the analysis.

6) Page 3, line 5: What do you mean by "good sensitivity"? Can you give a quantitative measure of what you mean by this?

7) Page 5, line 4: See Main Comment 3 above. How is the sensitivity calculated?

8) Page 5, lines 21-22: It is not easy to tell where Urmia and Tabriz are located. It would be good to label the locations of Yerevan, Urmia, and Tabriz on this map. Similarly, it would be good to have these labels on Figure 7.

9) Page 6, line 3: Can you please give a quantitative estimate for the improvement in the agreement?

10) Figure 3: It is difficult to see the details in this figure. Can you please enlarge the figure?

11) Figure 7: Please add the date of the observations to the figure caption.

---

## Author Comment (AC1) · 6 Feb 2019

We would like to thank reviewer 1, 2, and 3 for the constructive comments that aided us to improve our manuscript. In this post we provide our replies to the reviewer's comments. We provide a revised version of the manuscript, in which all changes are highlighted. Revised and added text is provided in blue. In our replies to the comment we provide line numbers, page numbers and figure numbers of the old version of the manuscript.

Please also note the supplement to this comment:

[Figure]

https://www.atmos-chem-phys-discuss.net/acp-2018-1185/acp-2018-1185-AC1-supplement.pdf
* * *

---

## Author Response (AR1)

**author comments on the manuscript acp-2018-1185, reviewer 1**

We would like to thank the reviewer for the constructive comments that aided us to improve our manuscript. In this document we provide our replies to the reviewer's comments, where the original comments made by the reviewer are numbered and typeset in italic and bold face font. Here line numbers, page numbers and figure numbers refer to the original version of the manuscript, if not stated differently. Additionally, the revised version of the manuscript is added.

1. ***On Page 5, line 5, the authors address the importance of a more realistic spatial distribution of pollution sources within the hot spot. Could the authors elaborate some more about this issue?***

   **adjusted** We changed the paragraph (p5,l2) from:

   "The remaining differences show that overall the simulation and the TROPOMI data agree well but with significant differences across the city. Although the model has a high CO sensitivity at the centre of Tehran (lower right panel) a trustworthy emission estimate would only be possible if the emission inventory gave a more realistic spatial distribution of pollution sources within the hot spot."
   to
   " Overall, the simulation and the TROPOMI data agree well, however still significant differences between the model and TROPOMI remain at the city of Tehran (lower left panel). This suggests that TROPOMI can sense pollution hot spots on sub-city scales that are not well reflected by the model. Even though the fit shows a high sensitivity for CO at the center of Tehran (lower right panel) a trustworthy emission estimate is only possible when those differences are reduced. To this end, the model calculation needs emission inventories that more realistically reflect the spatial distribution of the pollution sources within the hot spot including temporal resolution, which is not provided by the used EDGAR inventory. "

2. ***As this study is a highly valuable example for air pollution studies, the authors could provide further considerations while conducting a similar experiment in other cities, areas or countries and at different scales (city and sub-city).***

   **not adjusted**

   The reviewer is right. We are planning to apply this method to other cities and regions world wide. An extension of this study to including more regions and cities will require new WRF simulations that are computational demanding and a detailed study of the emission inventories available for the new regions. Hence, this goes beyond the scope of this publication which is intended to be an initial study. We will address emmission estimates from TROPOMI CO measurements at other regions in follow up studies. This is also added to the conclusions of the revised manuscript.

3. ***Page 5, line 33: difference***

   **adjusted** "differences" is changed to "difference"

4. ***Page 6, line 20: remove column***

   **adjusted** "column averaging column" is changed to "column averaging kernel"

**author comments on the manuscript acp-2018-1185, reviewer 2**

We would like to thank the reviewer for the constructive comments that aided us to improve our manuscript. In this document we provide our replies to the reviewer's comments. The original comments made by the reviewer are numbered and typeset in italic and bold face font. Following every comment we give our reply. Here line numbers, page numbers and figure numbers refer to the original version of the manuscript, if not stated differently. Additionally, the revised version of the manuscript is added.

1. ***The paper presents a nice example of an application of the use of TROPOMI data for model evaluation. However, it suffers from a somewhat too brief description of the research and the lack of details. Adding a more in-depth description of the applied methods and some more discussion would make the paper much more useful and also better readable.***

   **adjusted** Most of the methods is already published and we try to reduce the repetition of it. However, we hope that we can satisfy the reviewer by your changes to the manuscript below.

2. ***P 1, l 8: At larger scales = For background conditions?***

   **adjusted**

   We changed the sentence (p1,l8) from:

   "At large scales, TROPOMI CO agrees well with the WRF simulation with a mean difference of 5.7%." to
   " For background conditions, the WRF simulation agree well with TROPOMI CO, with a mean difference of 5.7%. "

3. ***P 1, l 15: since 13 October = on 13 October?***

   **corrected**

4. ***P 2, l 10: Please explain TCCON and NDACC.***

   **adjusted** We changed the sentence (p2,l10) from:

   " Already in the early phase of the mission a validation with TCCON and NDACC ground-based measurements showed compliance with the mission objectives on precision and accuracy (Borsdorff et al., 2018b). "

   to

   " Already in the early phase of the mission a validation with TCCON (Total Carbon Column Observing Network) and NDACC-IRWG (Network for the Detection of Atmospheric Composition Change - The Infrared Working Group) ground-based measurements showed compliance with the mission objectives on precision ($< 10\%$ ) and accuracy ($< 15\%$) (Borsdorff et al., 2018b). "

5. ***P 3, l 1: What is the true vertical profile in the context of this paper?***

   **adjusted** To clarify this we add the following sentence at p3, l7:

   " The data filtering is described in more detail by Borsdorff et al. (2018a). For the comparison of the TROPOMI CO data product with WRF model simulations, we apply the averaging kernel to the CO model profile $\vec{\rho}_{\mathrm{mod}}$ and compare $\mathbf{A}_{\mathrm{col}}\vec{\rho}_{\mathrm{mod}}$ directly to the TROPOMI CO column measurement $c_{\mathrm{ret}}$, where the averaging kernel accounts for the vertical sensitivity of the satellite measurement. "

6. ***P3, l 11-20: Please give some more details of the model setup. For example, which chemistry and/or tracer option is applied? How many tracers are considered? Which model resolution is applied? Why does the model domain extend that far towards the South?***

**not adjusted** Model details are already given in the submitted manuscript. The chemistry options chosen for WRF are described and referenced on p3,l10-11. The considered tracer runs are described in detail from p3, l21 - p4, l4. The spatial and vertical model resolution is stated on p3, l14-16.

Indeed, the domain extends more to the South then actually needed for this study. This was only to keep the possibility to study further sources that where finally not relevant.

7. *P 3, l 13: This particular topography is certainly not applied for the model simulations. So, for which purpose is it shown here?*

    **adjusted** We changed the sentence at p3, l12:

    " Figure 1 shows the topography of the area based on the 2000 Shuttle Radar Topography Mission (SRTM) data with an resolution of 15 arcsec (Farr et al., 2007). "
    to

    " The model domain includes regions of complex terrain of mountains and valeys as illustrated in Fig. 1, which affects regional weather processes. "
    Additionally we changed the figure caption of Fig.2 to:

    " Topological map of the model domain taken form the 2000 Shuttle Radar Topography Mission (SRTM) data with an resolution of 15 arcsec (Farr et al., 2007). An analogeous terrain height is used for the WRF simulation. The cities of Yerevan, Urmia, Tabriz, and Tehran with CO hot spots are marked in the map. "

8. *P 3, l 21: Please add some more information about the EDGAR emissions and how they are applied here.*

    **adjust** We changed the sentence at p3,l21 from:
    " Anthropogenic surface emissions of CO for 2010 are from the EDGARv4.2 emission inventory (Crippa et al., 2016). "
    to
    " Anthropogenic surface emissions of CO for 2010 are from the Emission Database for Global Atmospheric Research (EDGAR) version 4.2 (Crippa et al., 2016). The EDGAR inventory comprises global anthropogenic emissions based on publicly available data that can be used as input for atmospheric models. The emission used in this study are time invariant. "

9. *P 3, l22-32: The first line of the paragraph seems to be misplaced (or the second and third sentence should be part of the part of the description of the EDGAR emissions).*

    **adjusted** see our changes regarding the following point.

10. *P 3, l 27-32: At this point it is not clear why this is part of the description of WRF. Please explain this in the context with the WRF tracers.*

    **adjusted** We moved the description (p3, l22- p4,l4) to (p4,l19)

11. *P 4, l 1-2: Something seems to be missing in this sentence, please reword.*

    **adjusted** please see the following point

12. *P4, l 1: Please explain the meaning of prior CO sources.*

    **adjusted** We change the sentence p4,l1-2 from:

    " where the different coefficients of vector $\vec{\alpha}$ correspond to a scaling of the prior CO sources. The CO fields from the individual hot spots, $CO_{bkg}$ denotes the background field originating from the ECMWF-CAMS boundary conditions and $CO_{rest}$ summarises the contribution of the remaining EDGAR sources. "
    to
    " where the CO fields from the individual hot spots are named by the corresponding cities, $CO_{bkg}$ denotes the background field originating from the ECMWF-CAMS boundary conditions and $CO_{rest}$ summarises

the contribution of the remaining EDGAR sources. The different coefficients of vector $\vec{\alpha}$ describe a scaling of the corresponding CO fields and thus can be adjusted to fit the TROPOMI CO observations in the spatial domain of interest. "

13. **P 4, l 7-8: Does the interpolation of the WRF output in time and space mean that e.g. the top right of Figure 4 is a combination of outputs at different output times. If so, please mention this and/or give a description how this interpolation was made.**

**adjusted** We changed the sentence at p4,l7-8 from: " To compare the simulated CO fields with the TROPOMI CO column, we first interpolated the WRF data to the TROPOMI observation in time and space. "
to
" To compare the simulated CO fields with the TROPOMI CO columns, we first selected the CO field from the hourly WRF data, which is closest to the overpass time of TROPOMI and subsequently interpolate the model data to the geolocation of the individual TROPOMI ground pixels. "

14. **P 4, l 11 and l 31: at which time(s)?**

**adjusted** We changed the sentence (p4,l11) from:

" ... for an overpass on 18 December 2017 "
to
" ... for an overpass on 18 December 2017 9:37 UTC "

further we changed the sentence (p4,l31) from:

" ... WRF simulation over Tehran for 18 December 2017. "
to
" ... WRF simulation over Tehran for 18 December 2017 9:37 UTC "

For consistency we added the overpass times for the remaining orbits shown in this manuscript at p5,l18.

15. **P 4, l 17-18: This could be determined easily by inspection of the EMWF-CAMS fields.**

**adjusted** We changed the sentence on p4,l18 from:
" Furthermore, large scale deviations occur in the North-East of the model domain that are most probably caused by the ECMWF-CAMS side constraints.

" to

" Furthermore, large scale deviations occur in the North-East of the model domain that are caused by the ECMWF-CAMS side constraints since the same can be seen in the background CO tracer of WRF for the day shown in Fig. 2.
"

16. **P 4, l 24: Which ensemble?**

**adjusted** We changed the sentence (p4,l24) from:
" The statistics of the results are shown in Table 1 were we report the mean and the standard deviation of the ensemble as a robust error estimate. "
to

" The statistics of the results are shown in Table 1, which reports the mean and the standard deviation calculated from the emission estimates of the 1000 data samples as a robust error estimate. "

17. **P 4, l 13: Since emissions are usually higher during the daytime than during the night, the error due to temporally invariant emissions may depend on the time of the day. Therefore, information on the considered time of the day may be interesting.**

**adjusted** We added the time of the overpass following the previous comments of the reviewer. Furthermore, we addressed the time dependency of the emission source as an additional error source when we discussed the Tehran case.

18. **_P 5, l 15: This should also be mentioned already in section 2.2._**

    **adjusted** We changed the sentence section 2.2 at (p3,l21) from:

    " Anthropogenic surface emissions of CO for 2010 are from the EDGARv4.2 emission inventory (Crippa et al., 2016). "
    to

    " Anthropogenic surface emissions of CO for 2010 are from the Emission Database for Global Atmospheric Research (EDGAR) version 4.2 (Crippa et al., 2016). The EDGAR inventory comprises global anthropogenic emissions based on publicly available data that can be used as input for atmospheric models. The emission used in this study are time invariant. "

19. **_P 5, l 18-19: To what extent did these pollution events show up in the WRF simulations when the original emissions were applied?_**

    **adjusted** We changed the sentence p5, l23-26 from:

    " Using the EDGAR emissions, the large differences indicate that the enhancement cannot be explained by atmospheric transport as simulated by WRF and indicates an underestimation of the CO emissions in the EDGAR data. "
    to

    " Using the original EDGAR emissions without scaling, the large differences indicate that the enhancement cannot be explained by atmospheric transport as simulated by WRF and indicates an underestimation of the CO emissions in the EDGAR data. "

20. **_P 5, l 31: Does the WRF simulation really explain the observed CO enhancement?_**

    **adjusted** We change the sentence (p5,l31) from: " ...the WRF simulation can explain to a major extent the observed ..."
    to
    " the WRF simulation can reproduce to a major extent the observed "

21. **_P 6, l 29: The sentence The poor capability . . . is incomprehensible. Why is WRF not able to simulate at city scale. Is it due to the resolution, due to the emissions, or due to something else?_**

    **adjusted** We were not stating "poor capability" in the manuscript. In the contradiction we are impressed how good the agreement between WRF and TROPOMI already is. But of course there are always limiations that we addressed.

    We changed the sentence from:
    " Also, the capability of the WRF model to simulate CO on sub-city scales might be an additional source of uncertainty. "
    to
    " An additional source of uncertainty might be the capability of the WRF model to simulate CO on sub-city scales which is further limited by the availability of appropriate emission inventories but also the challenging task to model wind fields on this scale. "

22. **_P 7, l 2: What happens if the adapted emissions are applied for the entire episode? Is CO overestimated during observed episodes with moderate CO, or are the results still ok?_**

    **not adjusted**

    We understand the question of the referee to test if the derived emission estiamtes are in agreement with all observations of TROPOMI in this episode. However, we use all observations that are sensitive to the enhanced CO already in this study. Hence, consisitency can only be check by comparing the different emission estimates. This is done in this study. To answer this question it is necessary to extend the study for longer time periods of TROPOMI data. This will be done in a follow up study.

23. ***P 7, l 5: Please add some sentences on the potential of the method for other regions of the world.***

   **adjusted** We add the following sentence at p7, l6:

   " An interesting topic for follow up studies is to apply the method presented in this publication to other region world wide. For this it is important to restrict the analysis to clear-sky only scenes to ensure that the TROPOMI CO retrieval is sensitive for boundary layer pollution. Moreover, for bigger cities pollution on sub-city scales becomes more imporant and may need adjustment of the inversion approach. "

24. ***Data availability: The first link does not work and the second one is not really helpful in its current form. Please correct the links and add some explanation (if necessary, add supplementary material).***

   **todo create link** We corrected the links and changed the data availability section from: "

   The TROPOMI CO data set and the WRF simulations of this study are available for download at `ftp://ftp.sron.nl/pub/pub/DataProducts/TROPOMI_CO/`. The underlying data of the figures presented in this publication can be found at `ftp://ftp.sron.nl/open-access-data/`.
   "

   to

   " The TROPOMI CO data set and the WRF simulations of this study are available for download at `ftp://ftp.sron.nl/pub/pub/DataProducts/TROPOMI_CO/`. The underlying data of the figures presented in this publication can be found at `ftp://ftp.sron.nl/open-access-data-2/TROPOMI/tropomi/co/`.

   " The WRF simulation will be provided after publication of the manuscript.

25. ***Figure 3 includes numerous details, which are not necessary or enlarge the figure. Please mention the hour or time interval of the shown orbit.***

   **adjusted** We added the time of the overpass in the figure caption. Furthermore, we enlarged the figure.

26. ***Figure 4 and Figure 5: Please mention the hour, time interval or orbit(s) in the figure caption.***

   **adjusted** We added the time of the overpass in the figure caption.

27. ***Caption of Fig. 6: Please mention the region shown in this figure also in the caption.***

   **adjusted** We changed the figure caption from:

   " Sequence of high CO pollution events measured by TROPOMI on the 27th (top panel) November, 12th (middle panel), and 17th December 2017 (lower panel).
   "
   to
   " Sequence of high CO pollution events measured by TROPOMI on the 27th (top panel) November 9:31 UTC, 12th November 9:50 UTC (middle panel), and 17th December 2017 9:56 UTC (lower panel) above Armenia.
   "

28. ***Figures 6: Does this figure display the same region as the lower part of Fig. 2? If not: why? Please mention the time interval in the figure caption.***

   **adjusted** Yes it is the same region as the lower part of Fig. 2. Furthermore, we added the overpass times. We added the following sentence to the figure caption:
   " The figures show the same region as in the lower part of Fig. 2. "

29. ***Caption of Fig. 7: Please mention the date and the time (interval).***

**adjusted** We changed changed the figure caption from: " Comparison of TROPOMI CO and WRF (top row: WRF simulation, lower row difference TROPOMI - WRF). Left column: WRF run with EDGAR emissions. Middle column: WRF run with adjusted emissions for Yerevan, Tabriz and Urmia. Right column: WRF run with adjusted emission of Yerevan, Tabriz and Urmia and the traffic route near Yerevan.
"

to

" Comparison of TROPOMI CO and WRF (top row: WRF simulation, lower row difference TROPOMI - WRF) on the 17 December 2017 9:56 UTC above Armenia . Left column: WRF run with EDGAR emissions. Middle column: WRF run with adjusted emissions for Yerevan, Tabriz and Urmia. Right column: WRF run with adjusted emission of Yerevan, Tabriz and Urmia and the traffic route near Yerevan.
"

[Figure]

Figure 1: Relative difference between the WRF total column and the background CO tracer column (WRF-background)/background (left column) and total sensitivity of the WRF simualtion to CO emmsions along the road near Yerevan, and the cities Urmia and Tabriz for the 27th November 9:31 UTC (top row), 12th November 9:50 UTC (middle row), and 17th December 2017 9:56 UTC (lower row). For clearer presentation, we scaled the sensitivity fields of the road near Yerevan by 0.05, Urmia by 0.5, and Tabriz by 0.1.

[revised manuscript text omitted]